# AUTOGENDA: AUTOMATED GENERATIVE DATA AUGMENTATION FOR IMBALANCED CLASSIFICATIONS

## ABSTRACT

Data augmentation is an approach to increasing the training dataset size for deep learning using synthetic data. Recent advancements in image generative models have unleashed the potential of synthesizing high-quality images in data augmentation. However, real-life datasets commonly follow an imbalanced class distribution, where some classes have fewer samples than others. Image generation models may, therefore, struggle to synthesize diverse images for less common classes that lack richness and diversity. To address this, we introduce an automated generative data augmentation method, *AutoGenDA*, to extract and transfer label-invariant changes across data classes through image captions and text-guided generative models. We also propose an automated search strategy to optimize the data augmentation process for each data class, leading to better generalization. Our experiments demonstrate the effectiveness of *AutoGenDA* in various object classification datasets. We improve the standard data augmentation baselines by up to 4.9% on Pascal VOC, Caltech101, MS-COCO, and LVIS under multiple imbalanced classification settings.

## 1 INTRODUCTION

Conventional data augmentation methods involve applying pre-defined label-preserving image operations, such as horizontal flipping, random cropping, and rotation, to create multiple versions of the same image (Shorten & Khoshgoftaar, 2019). By providing ample examples of various variations, deep neural networks can learn to be resilient to nuisance transformations, such as color and geometric transformations that do not alter the class identity. This allows them to identify the objects despite these variations.

Modern deep image generative models can synthesize high-quality images given appropriate prompts. This capability allows the generation of a large number of realistic images, which could serve as artificial data to supplement image classification datasets without the need for costly data collection and labeling (Antoniou et al., 2017; Ramesh et al., 2021; Biswas et al., 2023; He et al., 2023; Trabucco et al., 2023). In data augmentation, generative models learn the underlying data distribution of each object class and generate multiple versions of the same object with slight differences (Antoniou et al., 2017; Trabucco et al., 2023). Compared to conventional data augmentation, generative approaches can capture more abstract and complex nuisance transformations, like pose, texture, and background changes, thus enabling greater diversity in the training data.

In practice, many real-life datasets follow an imbalanced class distribution, where some classes have a large number of samples while the remaining classes have very few. In cases of such imbalance, generative models may struggle to learn diverse variations for the underrepresented classes. For instance, in a "cat vs dog" classification task, if the "dog" category has only a few samples of a dog running on a grass field, a generative model may learn this specific data distribution and synthesize new images of dogs running with slight variations on a grass field. However, the classifier trained on this synthesized data may fail to generalize to other images of dogs in different scenes, like inside a house (Beery et al., 2018). To this end, we propose a novel generative approach that takes into consideration the nuisance variations in the dataset.

Our major assumption is that certain variances found in a target class are class-specific, while others are class-agnostic. For example, the species of cats in a cat image is a class-specific variance unique to the class category, while the background of the image is a class-agnostic variance. We believe

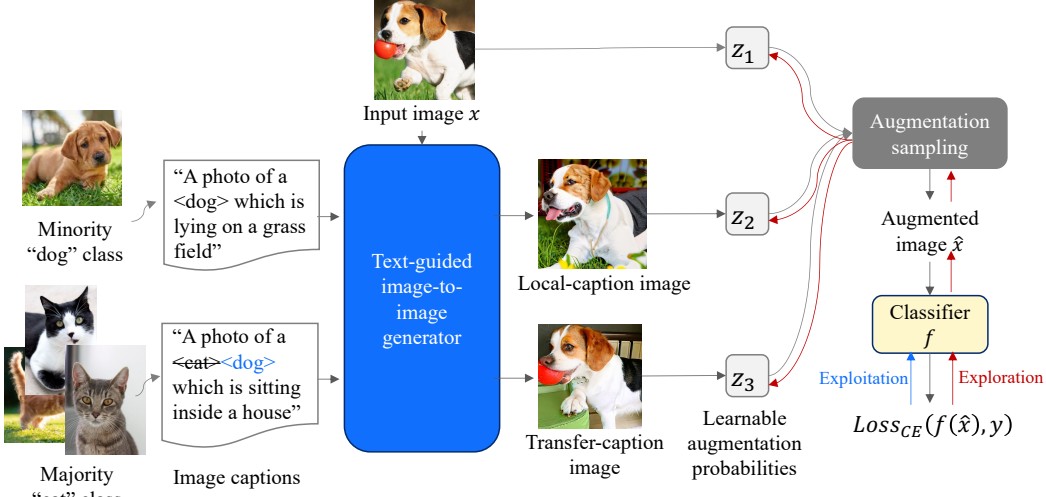

Figure 1: An overview of *AutoGenDA*. *AutoGenDA* extracts image captions from different classes and uses them as text prompts for editing images. These prompts are provided to a text-guided image-to-image generator to synthesize candidates of augmented images. Learnable augmentation probability parameters $z_1$, $z_2$, and $z_3$ are introduced to select the more effective augmented images to be included for classifier training. The exploitation pass updates the classifier $f$, and the exploration pass updates the augmentation parameters.

learning and transferring class-agnostic variances across similar classes in data augmentation can create a wider variety of images that are more effective for training classifiers under imbalanced class distributions. For example, a majority "cat" class contains images of cats in a house or on a grass field; a minority "dog" class contains only images of dogs on a grass field. By learning and transferring the scenery variances from the "cat" class to the "dog" class, a classifier can better handle unseen variations in less common categories. Our motivation is also supported by the finding from Zhou et al. (2022) that neural networks are not good at learning label-invariant transformations on the tail classes in long-tailed datasets.

Adopting generative data augmentation in imbalanced datasets poses at least two challenges. First, typical image generation models rely on large amounts of training data to learn good image representation. Extracting the variance properties from limited data and effectively transferring them across classes is not trivial. Second, in imbalanced datasets, each class may demand different numbers of augmented data and levels of diversity. In previous work, the decision on the amount or proportion of synthetic data to be included in the training dataset was mostly made manually, and the generated data were not guaranteed to be useful for the classification task. Ensuring the generated images are effective in our setting and automatically adapting the synthesized images to each class and dataset is a novel problem to solve.

Figure 1 shows an overview of *AutoGenDA*. At a high level, our method captures the class-specific and class-agnostic variances through image captions and integrates these differences into text-guided generation processes by using the captions as text prompts. We further propose a novel automated search framework to determine the optimal mixture of real and synthesized images to include for each class when fine-tuning a classifier on a dataset. To summarize, we make three major contributions:

- We propose a novel generative data augmentation method that learns and transfers class-specific and class-agnostic variances using image captions.

- We propose a novel automated search framework to optimize and adapt the generation process for each class and dataset.

- We demonstrate that our method outperforms other baselines on four object classification datasets under multiple imbalanced and low-shot settings.

## 2 RELATED WORKS

**Generative Data Augmentation.** The utilization of artificial data in data augmentation has gained traction due to advancements in image generation models. Early attempts leverage class-conditioned Generative Adversarial Networks (GANs) (Goodfellow et al., 2014) to generate augmented data that complements the training dataset (Antoniou et al., 2017; Tran et al., 2017; Bowles et al., 2018; Milz et al., 2018). Recently, diffusion models have been shown to generate images with higher quality compared to GANs (Rombach et al., 2022). He et al. (2023) applies SDEdit (Meng et al., 2022) on stable diffusion models to generate augmented data for training classifiers; DAFusion applies textual inversion (Gal et al., 2023) to fine-tune stable diffusion models for generating unseen visual concepts (Trabucco et al., 2023). These methods generally do not consider the variance information from different classes and may suffer from generating diverse images for minority classes in imbalanced datasets.

To learn nuisance transformations from multiple classes, GIT uses an image-to-image translation model to translate images from one class to another (Zhou et al., 2022). While this method is capable of learning lighting and color changes, it is unable to capture more complex alterations, such as object semantics and positions. In common object datasets, like CIFAR-10 and CIFAR-100, the method is reported to underperform compared to RandAugment (Cubuk et al., 2020), which uses a combination of conventional image transformations. The above-mentioned methods typically fix a pre-determined number or portion of augmented images for the training process without adapting the configurations for each class and dataset. For example, DAFusion and GIT augment half of the data, and GIT further restricts applying augmentations to low-data classes. To overcome these limitations, we introduce *AutoGenDA*. Unlike existing generative methods, *AutoGenDA* introduces a search framework that learns from classifier feedback and automatically adapts the augmentation process for each class.

**Automated Data Augmentation.** Conventional data augmentation applies label-preserving transformations to create new augmented data (Shorten & Khoshgoftaar, 2019). However, this approach relies heavily on expert knowledge or trial and error to determine the specific configurations and transformations to use. To address this shortcoming, Automated Data Augmentation (AutoDA) has been developed to learn the optimal policy for augmenting a target dataset (Cheung & Yeung, 2023). AutoAugment employs reinforcement learning to learn the probability and magnitude of applying multiple transformations to a dataset (Cubuk et al., 2019). Subsequent research has introduced more efficient search spaces and search methods to lessen the high computational demand of AutoAugment (Ho et al., 2019; Lim et al., 2019; Cubuk et al., 2020; Hataya et al., 2020; Li et al., 2020). Our work is also related to adaptive data augmentation, such as AdaAug (Cheung & Yeung, 2022), which learns adaptive augmentation strategies for each class and data instance. Previous AutoDA methods predominantly focus on searching for augmentation parameters for a set of image transformations but not for the generation process. To the best of our understanding, we are the first to study automated data augmentation in generative data augmentation.

## 3 METHODOLOGY

### 3.1 CAPTURING VARIANCE USING IMAGE CAPTIONS

Modeling data variances and transferring them across classes with limited data is challenging. We attempted to reference DAGAN (Antoniou et al., 2017) to train an image-conditioned GAN and use it to generate augmented data that is similar to the data from a class different from the conditioned images. However, the approach only works well for simple datasets like Omniglot and EMNIST, but fails to synthesize more complex data from object classification datasets, e.g., Caltech101 (Li et al., 2022). We also experimented with image-to-image translation (Huang et al., 2018) and style transfer (Karras et al., 2019) approaches to apply learned styles from one class to another. While these models are able to generate some valid images, the variations between the generated images are mostly limited to slight changes in color and lighting conditions. These findings are in line with previous research, indicating that these methods tend to struggle in an imbalanced setting, where some image classes lack richness and diversity (Patashnik et al., 2021).

To generate high-quality augmented images using limited data, we turn to large pre-trained text-guided generative models, which are capable of generating diverse, high-quality images by providing suitable text prompts (Meng et al., 2022; Rombach et al., 2022; He et al., 2023). While current

diffusion models show stunning generation results, they are sensitive to the provided prompts, which are usually crafted carefully through prompt engineering. Based on our assumption that some variance information within the dataset is class-agnostic, we propose to use image captions to capture the differences between data samples and integrate the captured variances into the text-guided generation process through text prompts.

Using image captions as text prompts in generation processes provides several advantages. First, these captions are automatically extracted from the target dataset, eliminating the need for expensive prompt engineering. The extracted captions describe the differences between data samples, providing important variance information for a classifier to learn. Additionally, textual captions are widely understood by any pre-trained text-guided generators, offering greater flexibility than using model-specific image or style representations to capture the variance between data samples.

We formally let $H(\boldsymbol{x}; y) : \mathcal{X} \to \mathcal{Q}$ be an image-captioning model that generates an image caption $q \in \mathcal{Q}$ given $\boldsymbol{x} \in \mathcal{X}$ as an input image and $y \in \mathcal{Y}$ as the class label of $\boldsymbol{x}$. Specifically, we use the prefix "a photo of a $\langle y \rangle$ which" to query the complete image caption from the model. The caption $q$ is later used to assist the generation of augmented images.

## 3.2 GENERATING NEW AUGMENTED IMAGES

Data augmentation aims to generate augmented data that is diverse and similar to the original data (Lopes et al., 2020). To preserve the image content, we use an image-to-image generative model that edits the original image instead of a text-to-image model that generates an image solely based on the input prompt. We use $G(\boldsymbol{x}; q) : \mathcal{X} \to \mathcal{X}$ to denote the generative model that takes the original image $\boldsymbol{x}$ as input and generates an augmented image conditioned on a text prompt $q$. To utilize the image variance found in one class, we sample an image $\boldsymbol{x}'$ where $y' = y$ and use the caption of $\boldsymbol{x}'$ as input prompt to guide the generation of the augmented data $\hat{\boldsymbol{x}}$ from $\boldsymbol{x}$ in Eq. (1). We refer to such type of augmented data that utilizes the image variance within the same class as *local-caption images*.

$$\hat{\boldsymbol{x}} = G(\boldsymbol{x}; q); \quad q = H(\boldsymbol{x}'; y'), \text{ where } y = y' \tag{1}$$

To create new data using variance information from a different class, we use a simple replace function $r(k; y) : \mathcal{Q} \to \mathcal{Q}$ to replace the class name in $k$ as $y$, where $k$ is the caption of an image from a different class. Let $\boldsymbol{x}''$ be an input image where $y'' \neq y$, the augmented data $\hat{\boldsymbol{x}}$ utilizing the caption of $\boldsymbol{x}''$ can be computed using Eq. (2). We call such type of augmented data that uses the image variance from another class as *transfer-caption images*.

$$\hat{\boldsymbol{x}} = G(\boldsymbol{x}; r(k, y)); \quad k = H(\boldsymbol{x}''; y''), \text{ where } y \neq y'' \tag{2}$$

The proposed method extracts data variances within and across different classes through image captions and uses them as editing instructions to create new images. While local captions are generally effective for describing images within the same class, this may not hold for cases involving using captions from another class as described in Eq. (2). For example, captions that describe a car may not be suitable for describing a dog. To tackle this issue, we employ a class-filtering mechanism. More precisely, we assume that captions from similar classes can be more easily applied to each other. For instance, using a caption for a cat to describe a dog is better than using a caption for a car. Therefore, the transfer-caption images are constrained to using captions from $m$ closest neighboring classes, measured by the cosine distance between the class embeddings. We use the notation $\texttt{neigh}(y) \in \mathcal{Y}^m$ to represent the set of $m$ closest classes to $y$, and set $m = 3$ by default. In our implementation, we utilize the pre-trained sentence transformer model from Reimers & Gurevych (2019) to obtain the class embeddings from the class names.

## 3.3 AUTOMATED SEARCH

Previous generative data augmentation typically generates a fixed number or portion of synthesized samples for all classes (Antoniou et al., 2017; Zhou et al., 2022; He et al., 2023; Trabucco et al., 2023). This heuristic approach does not guarantee the effectiveness of the synthetic data in improving classification generalization, particularly for imbalanced datasets where the optimal number of generated samples and the level of diversity may vary across different classes. In this section, we propose a new automated search framework that takes feedback from the classifier and optimizes the configuration for generative data augmentation for each individual data class. To the best of

our knowledge, our framework represents the first proposal for an automated search framework for generative data augmentation.

**Search Space**. Previous automated data augmentation work, like AutoAugment (Cubuk et al., 2019), learns the probability of applying pre-defined image operations to augment input images. We formulate the data augmentation process as a distribution learning problem to select the optimal mixture of different types of augmented data for each class. Specifically, the augmented image in *AutoGenDA* can be given as (1) an *identity image*, i.e., no augmentation is applied, (2) a *local-caption image*, or (3) a *transfer-caption* image. More formally, the augmentation process samples from a distribution $p(z; \alpha_y)$, where $\alpha_y \in \mathbb{R}^3$ is the learnable probability parameter specific to class $y$ and $z \in \{0, 1\}^3$, $z_1 + z_2 + z_3 = 1$ is the one-hot vector that indicates the selection of the augmentation from the three augmentation methods. Given a dataset with $C$ classes; the search space of $\alpha$ is $\mathbb{R}^{C \times 3}$. Let $q$ be an image caption sampled from class $y$ and $k$ be an image caption sampled from $\texttt{neigh}(y)$. The augmented image $\hat{x}$ is computed as:

$$\hat{x} = z_1 x + z_2 G(x; q) + z_3 G(x; r(k, y)), \quad z \sim p(z; \alpha_y). \quad (3)$$

**Differentiable Relaxation**. Inspired by Neural Architecture Search and Automated Data Augmentation (Dong & Yang, 2019; Li et al., 2020), we apply the Gumble Softmax trick (Jang et al., 2017) to make the discrete augmentation process continuous, thereby allowing $\alpha$ to be updated efficiently by stochastic gradient descent. More precisely, given $z_i$ as the $i$-th element in $z$, we relax $z_i$ with the learnable parameter $\alpha_y$ as:

$$z_i = \frac{\exp((\log \alpha_{y,i} + g_i)/\tau)}{\sum_{j=1}^3 \exp((\log \alpha_{y,j} + g_j)/\tau)}, \quad (4)$$

where $g_i = -\log(-\log(u_i))$ is the $i$-th Gumbel random variable, $u_i$ is a uniform random variable, and $\tau$ is the temperature parameter set to 1 by default in our implementation. For the inference time, the augmentation process uses the discretized form:

$$\hat{x} = z_1 x + z_2 G(x; q) + z_3 G(x; r(k, y)), \quad z = \text{one-hot}(\arg\max_i (\alpha_{y,i} + g_i)). \quad (5)$$

**Training.** *AutoGenDA* utilizes a two-stage training process. During the search stage, the probability parameter is updated through an iterative exploitation-exploration approach. In the exploitation step, a classifier $F_\theta$ parameterized by $\theta$ is trained to minimize the standard cross entropy loss on the augmented training dataset $\mathbb{D}_{\text{train}}$ using the discretized augmentation in Eq. (5). In the exploration step, $F_\theta$ is fixed and applied to the augmented validation dataset $\mathbb{D}_{\text{valid}}$ using the relaxed augmentation in Eq. (4). The probability parameter $\alpha$ is updated to minimize the cross entropy loss on the validation data. This adjustment of the probability parameter ensures a balanced mixture of real data, local-caption data, and transfer-caption data for each class to enhance generalization. It implicitly manages the diversity of data and the inclusion of local variance and transfer variance in the augmented data. Furthermore, this process can be regarded as a selection mechanism for indirectly choosing the more advantageous prompts when creating augmented images.

During the inference stage, we follow the exploitation step to fine-tune a classifier on the augmented training dataset using the learned probability parameter from the searching stage. The details of the search method are outlined in Algorithm 1. In Algorithm 1, $\texttt{mini-batch}$ is a batch sampling process, $\texttt{update}$ is a stochastic gradient descent procedure with a provided step size $\eta$ or $\zeta$, $\mathbb{Q}_y$ represents the set of captions of images from class $y$.

# 4 EXPERIMENTS AND RESULTS

## 4.1 EXPERIMENT SETUP

**Datasets.** We evaluate *AutoGenDA* on four image recognition datasets: PASCAL VOC (Everingham et al., 2015), Caltech101 (Li et al., 2022), MS-COCO (Lin et al., 2014), and LVIS (Gupta et al., 2019) under 16 imbalanced and low-shot settings. Caltech101, PASCAL VOC, and MS-COCO contain common real-life objects. Following the setup in DAFusion (Trabucco et al., 2023), we modify the PASCAL VOC and MS-COCO datasets for classification. Specifically, we select the images containing at least one object segmentation mask and assign the images the label corresponding to the object class with the largest area in the image. LVIS is a long-tail dataset derived from

---

**Algorithm 1** Training

---

**Input:** $\mathbb{D}_{\text{train}}, \mathbb{D}_{\text{valid}}, F_{\boldsymbol{\theta}}, G, H, \boldsymbol{\alpha}, \eta, \zeta$
1: $\mathbb{Q} \leftarrow \{q : q = H(\boldsymbol{x}, y), \text{where } (\boldsymbol{x}, y) \in \mathbb{D}_{\text{train}}\}$          ▷ Extract captions from $\mathbb{D}_{\text{train}}$
2: $\mathbb{Q}' \leftarrow \{q : q = H(\boldsymbol{x}, y), \text{where } (\boldsymbol{x}, y) \in \mathbb{D}_{\text{valid}}\}$       ▷ Extract captions from $\mathbb{D}_{\text{valid}}$
3: **while** not done **do**
4:      $\mathbb{A} \leftarrow \varnothing$
5:      **for each** $(\boldsymbol{x}, y) \in \texttt{mini-batch}(\mathbb{D}_{\text{train}})$ **do**             ▷ Exploitation
6:          $\bar{\mathbb{Q}} \leftarrow \bigcup_{y' \in \texttt{neigh}(y)} \mathbb{Q}_{y'}$         ▷ Find the neighbouring classes
7:          $q \overset{\text{R}}{\leftarrow} \mathbb{Q}_y, \ k \overset{\text{R}}{\leftarrow} \bar{\mathbb{Q}}$     ▷ Sample a caption from local class and transfer classes
8:          $\hat{x} \leftarrow \texttt{augment}(\boldsymbol{x}, y, q, k, \boldsymbol{\alpha})$        ▷ $\texttt{augment}$ computes Eq. (5)
9:          $\mathbb{A} \leftarrow \mathbb{A} \cup \{(\hat{\boldsymbol{x}}, y)\}$
10:      $\texttt{update}(\boldsymbol{\theta}; F_{\boldsymbol{\theta}}, \mathbb{A}, \eta)$                ▷ Update $\boldsymbol{\theta}$
11:      $\mathbb{A} \leftarrow \varnothing$
12:      **for each** $(\boldsymbol{x}, y) \in \texttt{mini-batch}(\mathbb{D}_{\text{valid}})$ **do**             ▷ Exploration
13:          $\bar{\mathbb{Q}} \leftarrow \bigcup_{y' \in \texttt{neigh}(y)} \mathbb{Q}'_{y'}$         ▷ Find the neighbouring classes
14:          $q \overset{\text{R}}{\leftarrow} \mathbb{Q}'_y, \ k \overset{\text{R}}{\leftarrow} \bar{\mathbb{Q}}$     ▷ Sample a caption from local class and transfer classes
15:          $\hat{x} \leftarrow \texttt{augment'}(\boldsymbol{x}, y, q, k, \boldsymbol{\alpha})$       ▷ $\texttt{augment'}$ computes Eq. (3) & (4)
16:          $\mathbb{A} \leftarrow \mathbb{A} \cup \{(\hat{\boldsymbol{x}}, y)\}$
17:      $\texttt{update}(\boldsymbol{\alpha}; F_{\boldsymbol{\theta}}, \mathbb{A}, \zeta)$                ▷ Update $\boldsymbol{\alpha}$

---

MS-COCO. The tail classes in LVIS typically contain rarer objects compared to the above-mentioned datasets (Gupta et al., 2019). We select a subset of the tail classes in LVIS to assess our method in classifying less common objects. To benchmark our method in an imbalanced setting, we set the minimum number of training samples for each class to be 2 and the maximum to be $c$, where $c$ is the number of samples in the class with the fewest samples from the original dataset. We then set the class frequency in the training set following an exponential distribution with the imbalance factors of 0.01, 0.1, 0.2, and 0.5 (ratio of the number of training examples between the most and least frequent classes). A lower imbalance factor indicates a more imbalanced class distribution. We also evaluate our method in balanced low-shot settings with 2, 4, 8, and 16 samples per class.

**Baselines.** We compare our method with four baselines. The simple baseline uses standard horizontal flipping, random cropping, and normalization. The RA baseline applies multiple random augmentations to an image using the default configuration in RandAugment (Cubuk et al., 2020). DAFusion (Trabucco et al., 2023) is a diffusion-based generative augmentation method that uses an image-to-image stable diffusion model with the fixed prompt template "A photo of $\langle y \rangle$" to generate an augmented image from class $y$. The GIT baseline uses a heuristic method to augment classes from minority classes with samples less than a pre-defined threshold (Zhou et al., 2022). We set the threshold to be 25% of the number of samples in the most frequent class. For a fair comparison, we replace the domain-adaptation generative model in the original GIT work with our image-to-image stable diffusion model. We use the same set of hyperparameters to fine-tune the stable diffusion model and generate the augmented images in all methods. To test the effectiveness of combining our method *AutoGenDA* with other data augmentation strategies, we provide the *AutoGenDA w/ RA* baseline, which applies RandAugment on top of the augmented data generated by *AutoGenDA*.

**Training.** We use Stable Diffusion v1.4 trained by Rombach et al. (2022) as the image generative model and apply SDEdit (Meng et al., 2022) for image-to-image translations. We follow DAFusion (Trabucco et al., 2023) to finetune the stable diffusion model on the target dataset using textual inversion (Gal et al., 2023) to capture novel visual concepts. For the image captioning model, we use the pre-trained BLIP2 from Li et al. (2023) and use the prefix "a photo of a $\langle y \rangle$ which" to query the complete image caption from the model. For the classification tasks, we employ a ResNet50 (He et al., 2016) model pre-trained on ImageNet. The classifier is fine-tuned on the target dataset for 50 epochs using a batch size of 32 and a learning rate of 0.0001. During the search phase, we split half of the dataset as the training data and the remaining half as the validation data. We use the same training protocol to update the classifier and stochastic gradient descent with a learning rate of 0.001 to update the probability parameters. For inference, we train the task model on the mixture of the full training data and the augmented data sampled using the probability parameter learned in the

Table 1: Comparsion of the test-set accuracy of *AutoGenDA* with other generative data augmentation baselines on imbalanced PASCAL VOC, Caltech101, MS-COCO, and LVIS datasets (*imb.* stands for the imbalance factor. A lower imbalance factor indicates a more imbalanced class distribution).

| Dataset | Simple | RA | DAFusion | GIT | *AutoGenDA* | *AutoGenDA w/ RA* |
|---|---|---|---|---|---|---|
| *PASCAL VOC* | | | | | | |
| *imb.* = 0.01 | $38.02_{\pm3.11}$ | $39.13_{\pm3.11}$ | $39.75_{\pm4.10}$ | $41.80_{\pm3.64}$ | $\mathbf{43.22}_{\pm3.82}$ | $43.79_{\pm3.79}$ |
| *imb.* = 0.1 | $61.00_{\pm2.45}$ | $62.43_{\pm2.13}$ | $62.45_{\pm2.14}$ | $64.25_{\pm2.23}$ | $\mathbf{65.02}_{\pm2.38}$ | $66.66_{\pm1.47}$ |
| *imb.* = 0.2 | $69.40_{\pm2.86}$ | $70.77_{\pm2.33}$ | $70.70_{\pm2.68}$ | $71.64_{\pm2.58}$ | $\mathbf{72.29}_{\pm2.03}$ | $73.61_{\pm1.01}$ |
| *imb.* = 0.5 | $77.68_{\pm1.29}$ | $\mathbf{78.81}_{\pm1.52}$ | $77.48_{\pm1.23}$ | $77.76_{\pm1.32}$ | $78.38_{\pm1.44}$ | $79.29_{\pm1.10}$ |
| *Caltech101* | | | | | | |
| *imb.* = 0.01 | $40.76_{\pm0.96}$ | $41.80_{\pm0.83}$ | $42.32_{\pm0.90}$ | $43.23_{\pm0.79}$ | $\mathbf{43.72}_{\pm0.81}$ | $45.16_{\pm1.01}$ |
| *imb.* = 0.1 | $69.78_{\pm1.36}$ | $72.04_{\pm1.41}$ | $72.02_{\pm1.37}$ | $72.86_{\pm1.19}$ | $\mathbf{74.13}_{\pm0.92}$ | $76.23_{\pm1.00}$ |
| *imb.* = 0.2 | $79.70_{\pm1.47}$ | $81.65_{\pm1.26}$ | $81.04_{\pm1.35}$ | $80.40_{\pm1.64}$ | $\mathbf{82.09}_{\pm0.75}$ | $83.76_{\pm0.48}$ |
| *imb.* = 0.5 | $85.16_{\pm0.85}$ | $\mathbf{87.73}_{\pm0.79}$ | $87.04_{\pm1.26}$ | $86.50_{\pm0.80}$ | $86.64_{\pm1.09}$ | $88.62_{\pm0.73}$ |
| *MS-COCO* | | | | | | |
| *imb.* = 0.01 | $22.00_{\pm1.42}$ | $23.39_{\pm1.32}$ | $23.22_{\pm1.82}$ | $\mathbf{25.19}_{\pm0.89}$ | $25.16_{\pm0.92}$ | $26.65_{\pm1.38}$ |
| *imb.* = 0.1 | $36.01_{\pm2.04}$ | $37.59_{\pm1.98}$ | $37.57_{\pm2.26}$ | $38.88_{\pm2.57}$ | $\mathbf{40.87}_{\pm2.36}$ | $42.19_{\pm1.80}$ |
| *imb.* = 0.2 | $43.38_{\pm2.54}$ | $44.85_{\pm2.35}$ | $45.19_{\pm1.97}$ | $44.73_{\pm2.27}$ | $\mathbf{47.63}_{\pm1.59}$ | $49.05_{\pm1.48}$ |
| *imb.* = 0.5 | $50.62_{\pm1.08}$ | $51.69_{\pm1.31}$ | $51.54_{\pm1.21}$ | $50.69_{\pm1.11}$ | $\mathbf{53.59}_{\pm0.97}$ | $55.25_{\pm1.28}$ |
| *LVIS* | | | | | | |
| *imb.* = 0.01 | $29.50_{\pm3.16}$ | $30.62_{\pm4.00}$ | $30.22_{\pm3.12}$ | $32.14_{\pm3.71}$ | $\mathbf{33.10}_{\pm3.00}$ | $33.49_{\pm4.24}$ |
| *imb.* = 0.1 | $49.89_{\pm2.85}$ | $50.56_{\pm2.61}$ | $49.62_{\pm3.60}$ | $52.98_{\pm3.52}$ | $\mathbf{53.36}_{\pm2.64}$ | $53.47_{\pm2.69}$ |
| *imb.* = 0.2 | $57.48_{\pm2.56}$ | $59.07_{\pm2.46}$ | $59.20_{\pm1.70}$ | $59.93_{\pm3.74}$ | $\mathbf{60.15}_{\pm2.71}$ | $61.06_{\pm2.20}$ |
| *imb.* = 0.5 | $62.78_{\pm2.48}$ | $63.31_{\pm1.48}$ | $64.63_{\pm1.94}$ | $62.88_{\pm1.75}$ | $\mathbf{64.85}_{\pm1.83}$ | $65.54_{\pm1.68}$ |

search phase. All the experiments are conducted on an NVIDIA RTX3090 GPU card. We repeat the experiments for eight random seeds and report the average test classification accuracy.

## 4.2 RESULTS

**Imbalanced Classification**. Table 1 shows the test accuracy on the four evaluation datasets with imbalance factors (*imb.*) of 0.01, 0.1, 0.2, and 0.5. As shown in the table, *AutoGenDA* outperforms other baselines in 13 out of 16 settings. In 2 of the remaining settings, where the datasets are less imbalanced (*imb.*=0.5), *AutoGenDA* performs comparably with the RandAugment baseline. In MS-COCO with *imb.* = 0.01, GIT slightly outperforms *AutoGenDA* by 0.03%. It appears that the heuristic setting of GIT works well in such a setting. However, in all other settings, the automated search framework in *AutoGenDA* finds a better way to augment the images. We observe that *AutoGenDA* provides a larger gain in more imbalanced situations. Specifically, it improves the simple baseline for 3% to 4.6%, 3.3% to 4.9%, 2.4% to 4.1%, and 0.7% to 3% for the imbalance factor equals 0.01, 0.1, 0.2, and 0.5, respectively. In Appendix B, we provide a statistical test to verify that *AutoGenDA* outperforms other baselines significantly. In addition, *AutoGenDA* can complement existing augmentation methods, like RandAugment. The *AutoGenDA w/ RA* baseline further improves the performance of *AutoGenDA* in all situations. This shows that *AutoGenDA* helps to learn a different set of variance information compared to those induced through traditional augmentation operations.

**Low-shot classification**. We also evaluate *AutoGenDA* on balanced low-shot settings. In Fig. 2, we compare the test accuracy of *AutoGenDA* with the Simple, DAFusion, and RandAugment baselines on the four evaluation datasets with 2, 4, 8, and 16 samples per class. The results show that *AutoGenDA* outperforms the other baselines in all settings. We also notice a larger improvement in lower-shot cases, like 2 samples per class, when compared with the cases with more samples per class. Although the gains are less obvious than those in the imbalanced setting, *AutoGenDA* can improve existing baselines in object classification tasks under low-shot settings.

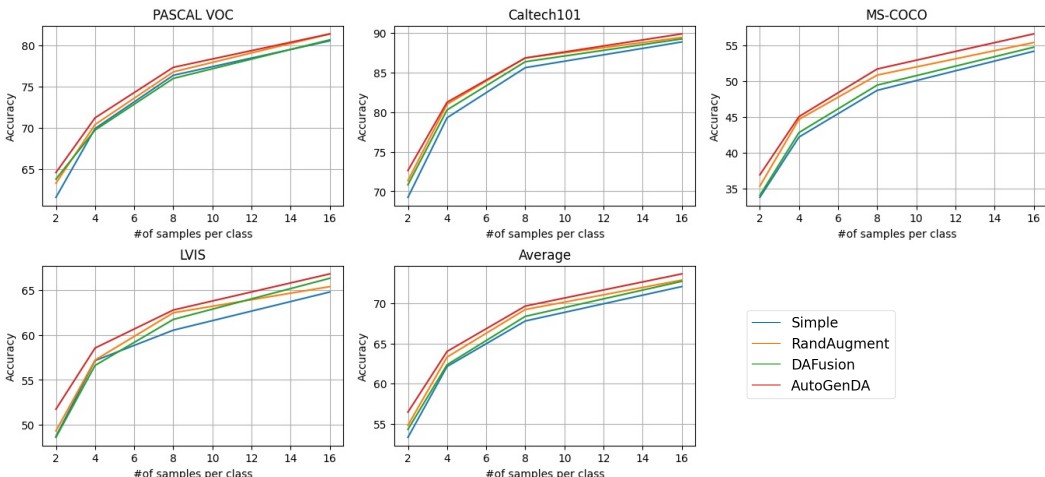

Figure 2: Comparison of the test accuracy of *AutoGenDA* and other baselines on the low-shot PASCAL VOC, Caltech101, MS-COCO, and LVIS datasets with 2, 4, 8, and 16 samples per class.

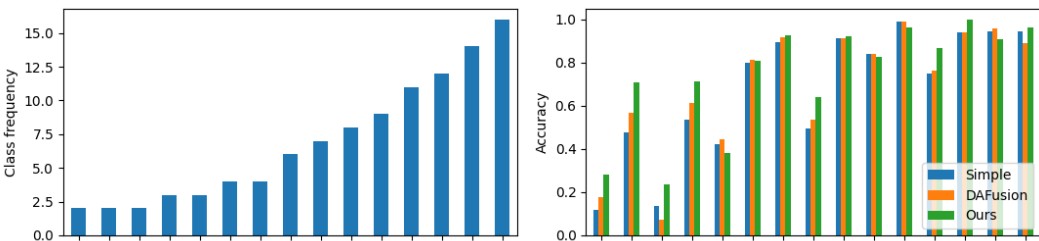

Figure 3: Illustration of the class frequency (left) and the corresponding test accuracy (right) for each class in the PASCAL VOC dataset with an imbalance factor of 0.1.

**Improvement in tail classes**. In this section, we investigate how *AutoGenDA* improves imbalanced classification tasks. Specifically, we examine the test accuracy of each class (shown in Fig. 3 right) and their corresponding class frequency (shown in Fig. 3 left). The visualization indicates that the improvements of *AutoGenDA* are more significant in the less frequent classes compared to the more frequent ones. Furthermore, the performance improvement of *AutoGenDA* in the minority classes surpasses that of the DAFusion baseline, which does not consider variance information from other classes. We have observed a similar pattern in other datasets as well. This evidence confirms that *AutoGenDA* is able to learn variance information from the more common classes and effectively apply it to the less common ones.

**Learned probability parameters.** We analyze the probability parameters learned from the PASCAL VOC dataset with 2 and 16 samples per class. Figure 4 presents the learned probability of sampling from the identity images (*identity-p*), the local-caption images (*samecls-p*), and the transfer-caption images (*othercls-p*). With 2 samples per class, the search method tends to explore the augmented images (green and orange bars) more than the original images (blue bar). When there are 16 samples per class, the search method focuses more on the original images than the augmented images. This can be explained by the fact that with less data, the classifier easily learns variance within the dataset and demands more varied data to improve generalization performance. Conversely, with more data, the classifier can learn diverse variance information from the real data. Additionally, we observe that the search method learns a different sampling distribution for each data class, indicating that *AutoGenDA* is capable of learning class-specific augmentation strategies.

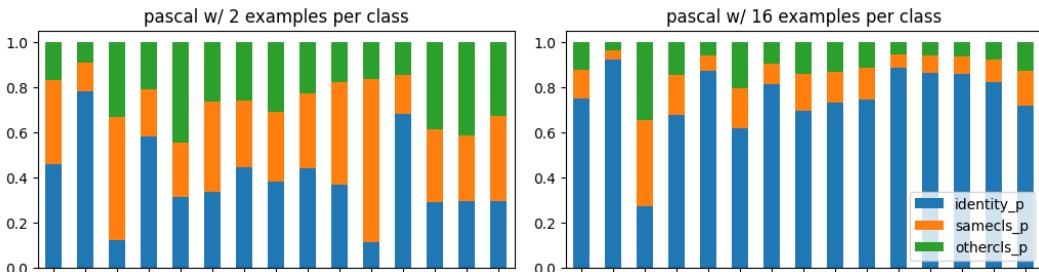

Figure 4: Comparision of the learned probability parameters on 15 sampled classes from the PASCAL VOC dataset with 2 and 16 samples per class.

## 4.3 DISCUSSION

**Qualitative analysis.** In Fig. 5, we show two examples of generated images using a simple prompt, a caption from the same class, and a caption from a neighbor class. The images synthesized using captions demonstrate more variations and align well with the texts. For example, we observe fog and buildings in the bus images and water and books in the elephant images. This supports the idea of using captions as prompts. Unlike image generation tasks, which focus on creating realistic image content, using generative models in data augmentation aims to enhance the test performance of classifiers trained on the augmented data. Although the captions may sometimes be less suitable for the target class, for instance, the image of an elephant sitting on books in Fig. 5, the search algorithm will auto-adjust the augmentation probability to sample less from these images if they do not contribute to the generalization of the classifier.

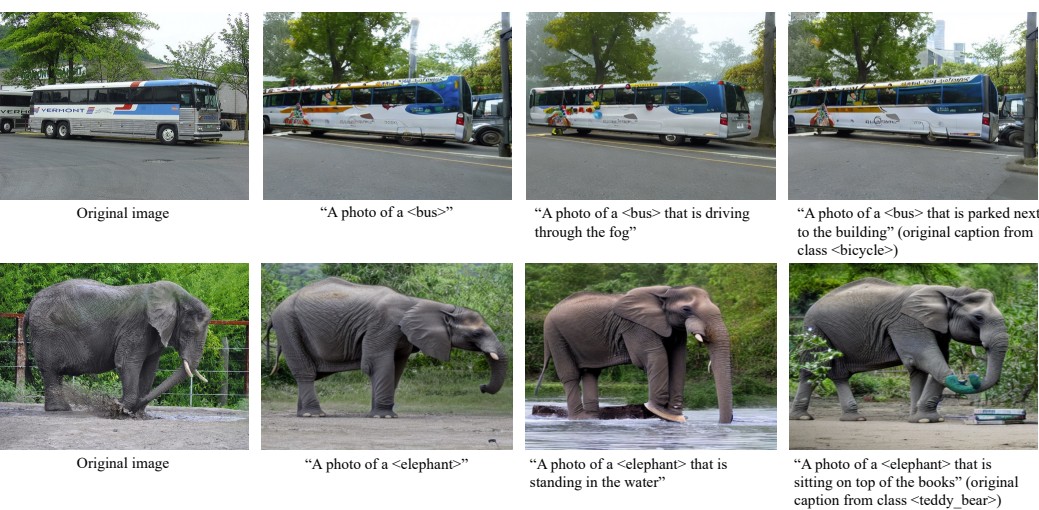

Figure 5: Illustration of the augmented data synthesized using different prompts.

**Limitations.** Our approach relies on a pre-trained image captioning model to generate descriptions that highlight the dissimilarities between input images. However, the approach may be less effective if the target domain drastically differs from the domain on which the model is trained. Additionally, our method involves an extra search phase to determine the augmentation parameter before training the task network. If computational power is a concern, an alternative method is the search-free *AutoGenDA* baseline introduced in the ablation section. The search-free method uniformly samples the augmented images to train the classifier network. Our study demonstrates that the search-free baseline yields promising results when compared to existing baselines.

## 5 CONCLUSION

In this paper, we introduce an automated generative data augmentation method called *AutoGenDA*. *AutoGenDA* learns variance information within and across classes using image captions and adapts the generative process to each data class and new dataset by learning the best combination of real data and augmented samples for each class. Through qualitative and quantitative analysis, we demonstrate the effectiveness of *AutoGenDA* on four classification datasets with different imbalanced and low-shot settings. Our search method is not dependent on any specific model or architecture. This provides great flexibility in adapting contemporary generative and image captioning models, which could further enhance its performance in imbalanced and low-shot classification.

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

## A    CONVERGENCE ANALYSIS

In Fig. 6, we track the evolution of the learnable probability parameter $z$ during training on PASCAL VOC. From left to right, the plots in Fig. 6 represent the learnable parameter $z_1$ for sampling an identity image, the parameter $z_2$ for sampling local-caption images, and the parameter $z_3$ for sampling transfer-caption images. Each line in the figure corresponds to a learnable parameter for a specific class. All the learnable parameters are initialized with the same value and updated to maximize the validation performance. As the training progresses, the probability parameters converge to some fixed points and exhibit a trend of convergence.

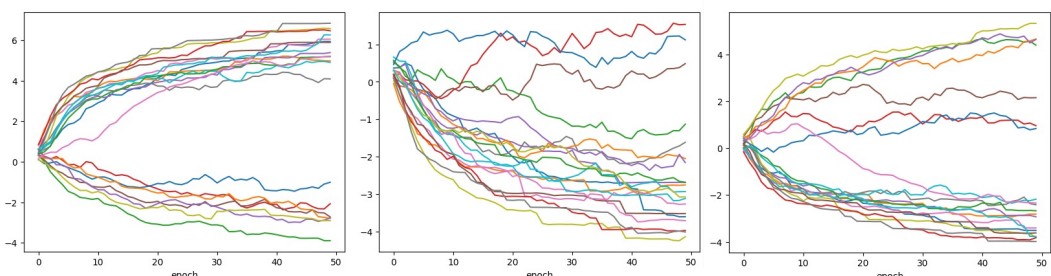

Figure 6: Illustration of the convergence of the learned augmentation probabilities on PASCAL VOC. The left figure shows the parameter $z_1$ for sampling an identity image; the middle figure shows the parameter $z_2$ for sampling a local-caption image; the right figure shows the parameter $z_3$ for sampling a transfer-caption image.

## B  STATISTICAL ANALYSIS

In this section, we use hypothesis testing methods to provide statistical support for our experimental results. Specifically, we follow the statistical comparisons of classifiers over multiple datasets suggested by Demšar (2006). We use the Friedman test to detect the statistical differences in the ranks among the baselines evaluated on the sixteen imbalanced datasets. We then use the Holm-Bonferroni post-hoc test to validate the significance of our improvements over each baseline. The p-value of the Friedman test is computed as 9.992e-16. Thus, at a significance level of 0.05, we reject the null hypothesis, which states that all methods perform equally.

Following the rejection, we conduct the Holm-Bonferroni post-hoc test to compare a control method, in our case *AutoGenDA*, with other baselines. In Table 2, we report the p-values for each comparison. From the results, we can reject the null hypothesis for all baselines, as the p-values are smaller than the adjusted $\alpha$'s. Therefore, we conclude that *AutoGenDA* has a significantly better ranking over all the four baselines on the tested datasets.

Table 2: Statistical test results with $\alpha = 0.05$.

| $i$ | Method | p-value | $\alpha/i$ | is significant? |
|---|---|---|---|---|
| 1 | simple | 1.685043e-10 | 0.0125 | Yes |
| 2 | DAFusion | 1.709823e-04 | 0.0167 | Yes |
| 3 | RandAugment | 5.077373e-03 | 0.0250 | Yes |
| 4 | GIT | 1.012654e-02 | 0.0500 | Yes |

## C  PRELIMINARY ATTEMPTS IN USING GANS AND IMAGE STYLE TO AUGMENT IMBALANCED DATASETS

### C.1  DAGAN

In our initial experiments, we attempted to utilize Generative Adversarial Networks (GANs) for generating augmented images. Inspired by DAGAN, we utilize an input-conditioned GAN to create images that are similar to but different from the input image. Specifically, we use an image encoder to map the input image to a latent representation. The latent representation is then combined with a transformed noise and decoded by an image decoder to produce the augmented image. A discriminator is employed to distinguish the real sample pairs and fake sample pairs. The real sample pairs comprise the input image and another image from the class as the input image, whereas the fake sample pairs are composed of the input image and the image generated by the decoder.

In the Omniglot dataset, the DAGAN approach can generate visually appealing augmented characters that are similar yet subtly different from the original image, as demonstrated in Fig. 7. However, when applied to more complex object datasets like CIFAR-10, we observed that the approach is unable to produce photorealistic images, as depicted in Fig. 8. Despite experimenting with a larger image encoder, image decoder, latent representation, and longer training iterations, the results remained similar.

### C.2  MUNIT

MUNIT is a multimodal unsupervised image-to-image translation network, which is capable of transforming a given input image into various outputs in different styles using the style codes learned from the data. In GIT (Zhou et al., 2022), MUNIT is modified to learn the translation between different classes instead of between two different domains. More specifically, MUNIT decomposes the latent space of images into a content space and style space and introduces reconstruction and cycle-consistency losses to generate augmented images that preserve the image content but in a different style. With the use of MUNIT, we are able to generate realistic images in CIFAR-10, as illustrated in Fig. 9. However, we have noted that the variations in the generated images primarily involve minor changes in color and lighting conditions. This discovery aligns with previous research, which suggests that the method faces challenges in an imbalanced setting, where certain image classes lack richness and diversity.

Original Images

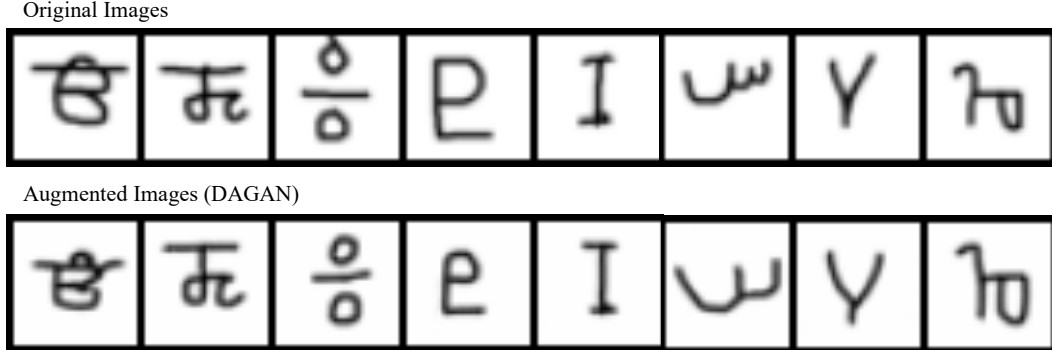

Augmented Images (DAGAN)

Figure 7: Illustration of the augmented results on Omniglot dataset using a modified DAGAN.

Original Images

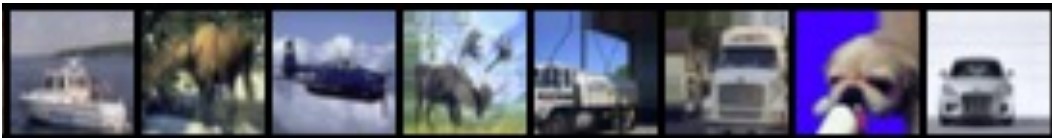

Augmented Images (DAGAN)

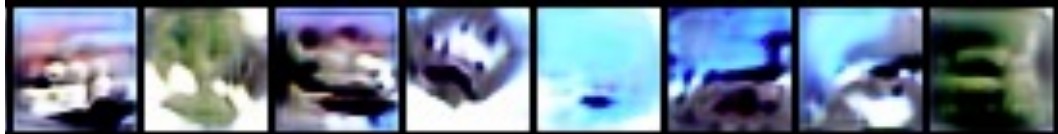

Figure 8: Illustration of the augmented results on CIFAR-10 dataset using a modified DAGAN.

## D  ABLATION STUDY

NEW

In this section, we investigate the effectiveness of different components in *AutoGenDA* by removing each of them from the method. The search-free baseline assigns a fixed probability parameter to uniformly sample from identity, local-caption, and transfer-caption images. For the second baseline, we only use local-caption images as the augmented data. For the third baseline, we only use transfer-caption images as the augmented data. In Table 3, we present the test accuracy of the ablation baselines on the four evaluation datasets with imbalanced factors of 0.01, 0.1, 0.2, and 0.5. The study shows that the results are compromised if we remove the images generated by captions from the same class or captions from other classes. This validates our motivation in learning class-specific variance through local-caption images and class-agnostic variance through transfer-caption images. We also verify that the search algorithm helps optimize the augmentation process to provide a better mix of augmented data.

We also conduct the Friedman test to detect the statistical differences in the ranks among the ablation baselines evaluated on the sixteen imbalanced datasets. We then use the Holm-Bonferroni post-hoc test to validate the significance of our method over each ablation baseline. The p-value of the Friedman test is computed as 1.11e-16. Thus, at a significance level of 0.05, we reject the null hypothesis, which states that all ablation baselines perform equally.

Following the rejection, we conduct the Holm-Bonferroni post-hoc test to compare a control method, in our case *AutoGenDA*, with other ablation baselines. In Table 4, we report the p-values for each comparison. From the results, we can reject the null hypothesis for all ablation baselines, as the p-values are smaller than the adjusted $\alpha$'s. Therefore, we conclude that *AutoGenDA* has a significantly better ranking over all the four ablation baselines on the tested datasets.

Original Images

Augmented Images (MUNIT)

Figure 9: Illustration of the augmented results on CIFAR-10 dataset using a modified MUNIT model.

Table 3: Ablation study on the use of automated search, images generated without the captions from the same class and those generated without the captions from other class. The reported results are the average test accuracy on the four datasets with different imbalanced settings.

| Dataset | Simple | Search-free | w/o transfer-caption | w/o local-caption | AutoGenDA |
|---|---|---|---|---|---|
| *PASCAL VOC* | | | | | |
| *imb.* = 0.01 | $38.02_{\pm3.11}$ | $42.09_{\pm4.50}$ | $42.41_{\pm3.74}$ | $41.89_{\pm4.25}$ | $43.22_{\pm3.82}$ |
| *imb.* = 0.1 | $61.00_{\pm2.45}$ | $64.15_{\pm2.13}$ | $63.95_{\pm2.32}$ | $63.47_{\pm1.85}$ | $65.02_{\pm2.38}$ |
| *imb.* = 0.2 | $69.40_{\pm2.86}$ | $71.66_{\pm1.36}$ | $71.54_{\pm1.38}$ | $71.11_{\pm1.71}$ | $72.29_{\pm2.03}$ |
| *imb.* = 0.5 | $77.68_{\pm1.29}$ | $77.98_{\pm0.95}$ | $78.09_{\pm0.87}$ | $78.12_{\pm1.02}$ | $78.38_{\pm1.44}$ |
| *Caltech101* | | | | | |
| *imb.* = 0.01 | $40.76_{\pm0.96}$ | $43.51_{\pm0.88}$ | $43.36_{\pm1.00}$ | $43.17_{\pm0.94}$ | $43.72_{\pm0.81}$ |
| *imb.* = 0.1 | $69.78_{\pm1.36}$ | $73.71_{\pm1.47}$ | $73.78_{\pm0.87}$ | $73.86_{\pm0.96}$ | $74.13_{\pm0.92}$ |
| *imb.* = 0.2 | $79.70_{\pm1.47}$ | $82.07_{\pm0.75}$ | $81.77_{\pm0.78}$ | $81.73_{\pm0.83}$ | $82.09_{\pm0.75}$ |
| *imb.* = 0.5 | $85.16_{\pm0.85}$ | $87.44_{\pm0.79}$ | $87.32_{\pm1.26}$ | $87.45_{\pm0.80}$ | $86.64_{\pm1.09}$ |
| *MS-COCO* | | | | | |
| *imb.* = 0.01 | $22.00_{\pm1.42}$ | $24.85_{\pm1.87}$ | $24.68_{\pm0.83}$ | $24.88_{\pm1.23}$ | $25.16_{\pm0.92}$ |
| *imb.* = 0.1 | $36.01_{\pm2.04}$ | $39.25_{\pm2.44}$ | $40.52_{\pm1.98}$ | $40.31_{\pm2.43}$ | $40.87_{\pm2.36}$ |
| *imb.* = 0.2 | $43.38_{\pm2.54}$ | $47.47_{\pm1.44}$ | $46.70_{\pm2.17}$ | $46.02_{\pm2.15}$ | $47.63_{\pm1.59}$ |
| *imb.* = 0.5 | $50.62_{\pm1.08}$ | $52.64_{\pm1.13}$ | $53.25_{\pm1.43}$ | $53.00_{\pm1.58}$ | $53.59_{\pm0.97}$ |
| *LVIS* | | | | | |
| *imb.* = 0.01 | $29.50_{\pm3.16}$ | $32.27_{\pm4.54}$ | $31.57_{\pm4.12}$ | $32.10_{\pm2.81}$ | $33.10_{\pm3.00}$ |
| *imb.* = 0.1 | $49.89_{\pm2.85}$ | $51.34_{\pm3.07}$ | $52.34_{\pm3.06}$ | $51.12_{\pm2.56}$ | $53.36_{\pm2.64}$ |
| *imb.* = 0.2 | $57.48_{\pm2.56}$ | $59.84_{\pm1.62}$ | $59.20_{\pm2.34}$ | $60.01_{\pm1.61}$ | $60.15_{\pm2.71}$ |
| *imb.* = 0.5 | $62.78_{\pm2.48}$ | $64.43_{\pm1.50}$ | $64.15_{\pm2.15}$ | $64.75_{\pm1.75}$ | $64.85_{\pm1.83}$ |

# E  EXPERIMENTS ON CIFAR-10 AND CIFAR-100

NEW

We add additional experiments on long-tail CIFAR-10-LT and CIFAR-100-LT datasets. We set the imbalance factor to 0.01 and compared the test accuracy among the baselines. FGDS is a feedback-guided text-to-image generation method proposed by Askari-Hemmat et al. (2023). It utilizes the feedback from a classifier pre-trained on the long-tail dataset and encourages the generated images to be more similar to the images from the given dataset. As shown in Table 5, *AutoGenDA* slightly leads the other baselines on the two tested datasets. The training protocol is same as the main experiment described in the main content.

Table 4: Statistical test results for the ablation study with confidence level $\alpha = 0.05$.

| $i$ | Method | p-value | $\alpha/i$ | is significant? |
|---|---|---|---|---|
| 1 | simple | 7.492673e-12 | 0.0125 | Yes |
| 2 | w/o transfer-caption | 1.039858e-03 | 0.0167 | Yes |
| 3 | w/o local-caption | 1.056898e-03 | 0.0250 | Yes |
| 4 | Search-free | 1.185697e-03 | 0.0500 | Yes |

Table 5: Comparsion of the test-set accuracy of *AutoGenDA* with other data augmentation baselines on imbalanced CIFAR-10-LT and CIFAR-100-LT.

| Dataset | Simple | RA | DAFusion | GIT | FGDS | AutoGenDA |
|---|---|---|---|---|---|---|
| CIFAR-10-LT | $56.80_{\pm1.21}$ | $59.49_{\pm3.13}$ | $55.49_{\pm1.97}$ | $59.81_{\pm1.77}$ | $59.29_{\pm1.40}$ | $60.08_{\pm1.43}$ |
| CIFAR-100-LT | $28.46_{\pm2.45}$ | $31.06_{\pm2.53}$ | $29.30_{\pm1.75}$ | $32.54_{\pm2.52}$ | $32.51_{\pm2.41}$ | $33.11_{\pm2.55}$ |

## F COMPUTATION OVERHEAD

NEW

To compare the computation overhead, we report the GPU hours needed to complete the augmentation processes for the CIFAR-10-LT dataset among the tested baselines. All the experiments are conducted on a single NVIDIA RTX3090 GPU card. The computation effort is mainly dominated by the image-generating process. The search effort for AutoGenDA only occupies a small portion of the total computation time.

Table 6: Comparision of the computation overhead.

| Process | Textual Inversion | Image Captioning | Img2Img Generation | Feedback-guided Text2Img Generation | Automated Search | Classifier Training | Total GPU hour |
|---|---|---|---|---|---|---|---|
| GPU-hours | 1.3 | 2.5 | 4.9 | 13.7 | 1.9 | 0.9 | - |
| DAFusion | ✓ | | ✓ | | | ✓ | 7.1 |
| GIT | ✓ | ✓ | ✓ | | | ✓ | 9.6 |
| FGDS | | | | ✓ | | ✓ | 14.6 |
| Ours | ✓ | ✓ | ✓ | | ✓ | ✓ | 11.5 |
| Ours (search-free) | ✓ | ✓ | ✓ | | | ✓ | 9.6 |

## G ANALYSIS ON THE LEARNED SAMPLING PARAMETERS

NEW

In Fig. 10, we show the averaged augmentation probabilities over all classes for the PASCAL VOC, Caltech, COCO, and LVIS datasets with 2, 4, 8, and 16 shots of data.

The learned probabilities for identity images show an increasing trend, while the learned probabilities for local-caption and transfer-caption images show a decreasing trend. This can be explained by the fact that with less data, the classifier easily learns variance within the dataset and demands more varied data to improve generalization performance. Conversely, with more data, the classifier can learn diverse variance information from the real data, thereby assigning a larger probability to sample an identity image.

## H MORE VISUAL EXAMPLES

NEW

We provide more examples of the generated images using *AutoGenDA* for the PASCAL VOC and MS-COCO datasets in Fig. 11 and Fig. 12, respectively.

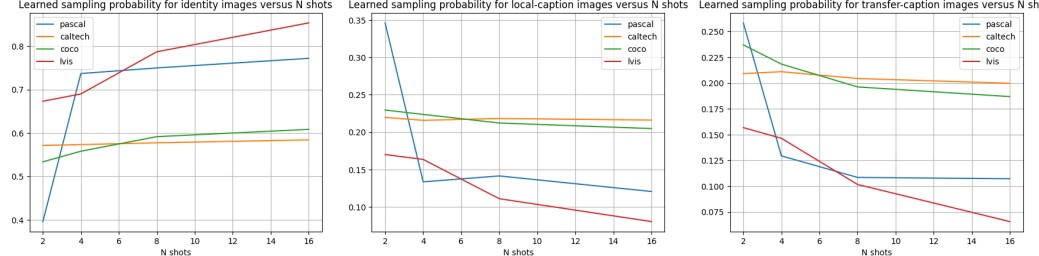

Figure 10: Illustration of the learned augmentation probabilites versus the number of shots on PASCAL VOC. The left figure shows the parameter $z_1$ for sampling an identity image; the middle figure shows the parameter $z_2$ for sampling a local-caption image; the right figure shows the parameter $z_3$ for sampling a transfer-caption image.

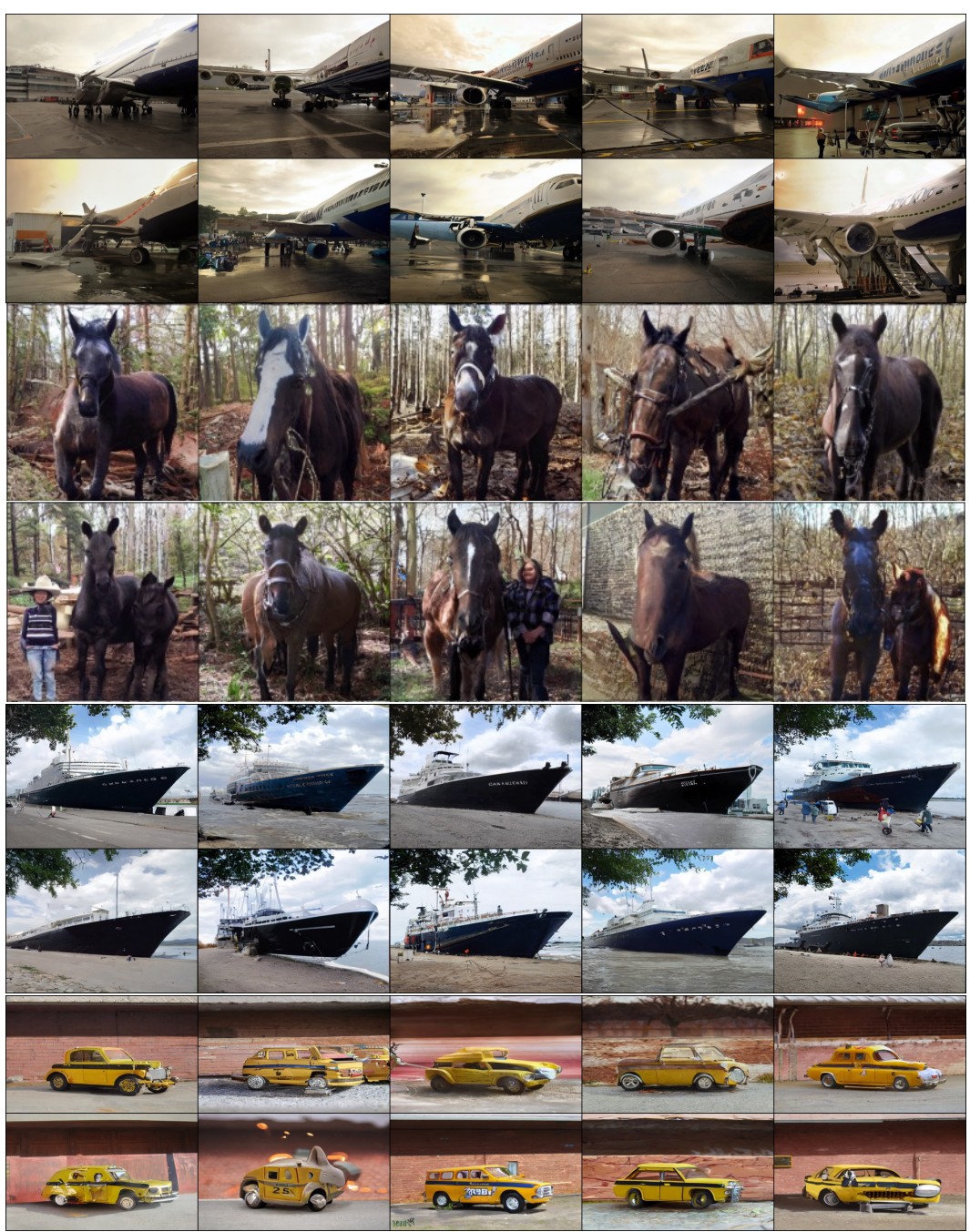

Figure 11: Illustration of the augmented images from the PASCAL VOC dataset.

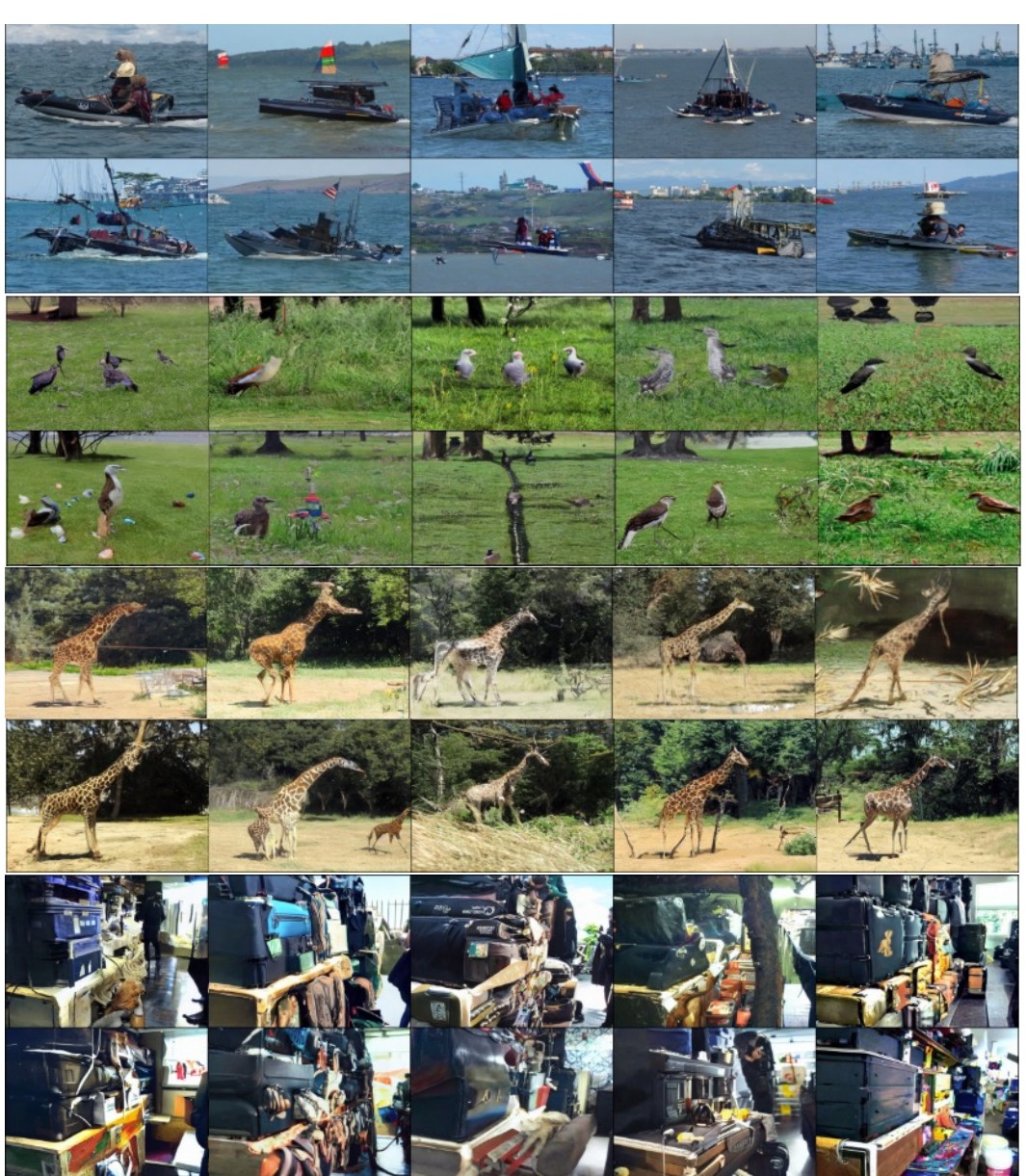

Figure 12: Illustration of the augmented images from the MS-COCO dataset.

