# OpenReview forum: "AutoGenDA: Automated Generative Data Augmentation for Imbalanced Classifications"
_ICLR.cc/2025/Conference — Submitted to ICLR 2025_

### Official Review · Reviewer_vKB1 · 2024-10-20

**Soundness:** 2
**Presentation:** 2
**Contribution:** 2
**Rating:** 3
**Confidence:** 4

**Summary:**

The paper introduces AutoGenDA, an automated generative data augmentation method aimed at improving deep learning performance on imbalanced datasets. AutoGenDA uses image captions and text-guided generative models to extract and transfer label-invariant changes across different classes to enhance diversity in underrepresented categories. An automated search strategy is proposed to optimize the augmentation process for each class, improving model generalization. The authors claim that AutoGenDA improves baseline data augmentation methods by up to 4.9% on several object classification datasets, including Pascal VOC, Caltech101, MS-COCO, and LVIS.

**Strengths:**

1. The problem of learning from long-tailed data is important in practice.
2. The idea is natural.

**Weaknesses:**

1. The basic assumption may be not fully correct. I think the text-guided image-to-image generator should also be trained with long-tailed data.
2. Experiments are not sufficient. Results on ImageNet-LT and iNaturalist should be provided.
3. Many important baselines are not considered. The authors are suggested to compare their proposed method with the baselines considered in [*1, *2]

[*1] Generalized Parametric Contrastive Learning, TPAMI
[*2] Probabilistic Contrastive Learning for Long-Tailed Visual Recognition, TPAMI

**Questions:**

See Weaknesses.

---

> ### Author Response · Authors · 2024-11-23
>
> [W1] __Fine-tuning.__ Thanks for your comment for training the generator with long-tailed data. We are afraid that the reviewer may miss some parts of our work. We would like to clarify that the generator of our method is finetuned using the textual inversion technique on the long-tailed datasets (in P.6, line 315).
>
> [W2/W3] __More baselines and datasets__. Thanks for your suggestion. Under the limited rebuttal period, we apologize for not being able to include all datasets and baselines suggested by every reviewer. We add the baseline Feedback-Guided Data Synthesis (FGDS) suggested by Reviewer AgGB as it is a recent work that also focuses on imbalanced classification, which is more related to our work. We also follow the suggestion by Reviewer AgGB to add the CIFAR-10-LT and CIFAR-100-LT experiments. The new results are illustrated in Table 5 of Appendix E. Our method slightly leads the other baselines on the two new datasets.

---

### Official Review · Reviewer_AgGB · 2024-10-31

**Soundness:** 2
**Presentation:** 2
**Contribution:** 1
**Rating:** 5
**Confidence:** 5

**Summary:**

This paper focuses on leveraging generative models as tools to produce data augmentations that can improve a classifier's accuracy for imbalance and low-shot classification. The authors use BLIP2 to generate captions from images and StableDiffusion to generate images from captions. The authors suggest to use the captions generated by BLIP2 using the real images that are in the majority group as augmented prompt that can be use to generate more example from the minority groups. To do so, they just replace the class label from the majority group's image caption by the class label of the minority group and use StableDiffusion to generate the corresponding image. This approach allow the authors to increase the number of examples in the minority group which improve performances on imbalance classification tasks. To avoid matching class labels with captions that could be not compatible (like "a car which is sitting in the house), the authors leverage class name embeddings to only apply "caption transfer" between class that are closed by in the embedding space.

**Strengths:**

- The idea is well explained and simple to understand. It make sense to try to leverage the additional diversity we have in the majority groups to augment the number of classes in the minority groups.
- Authors show good improvements over the DAFusion baseline.
- I appreciate that the authors took the time to describe their preliminary results that did not work in the appendix.

**Weaknesses:**

- Lack of relevant related works. The authors claim that they are the first to study automated data augmentation with generative models which clearly show they are not aware that there is an entire research field on this question. I added few references at the end of this comment ( 1) 2) 3) 4) )
- This lack of relevant related work induce a lack of relevant baselines. I would have expected to see comparisons between AutoGenDA and methods like Fill-Up, Diffuse-Mix or the others.
- Lack of novelty. If we look at the works I just listed, I am not convinced that the method presented by the authors is any better than the current existing methods for data augmentation using synthetic data.
- Authors did not use any of the traditional datasets use to evaluate imbalance classification (like CIFAR-10-LT, CIFAR100-LT, Places-LT ,ImageNet-LT, Inat18, NICO++).
- The ablation study does not show for now statistically significant improvement by using AutoGenDA (~0.5% d'accuracy and the authors does not present what is the variance or the dataset they are using for the ablation).

1) Fill-up: Balancing long-tailed data with generative models, Shin et al 2023
2) Feedback-guided Data Synthesis for Imbalanced Classification Askari-Hemmat et al 2023
3) DreamDA: Generative Data Augmentation with Diffusion Models Fu et al. 2024
4) DiffuseMix: Label-Preserving Data Augmentation with Diffusion Models Islam et al. 2024

**Questions:**

- Since the image generation will mostly affect the background for the minority classes, do you have any insights wether the method might just improve robustness with respect to the background? There is the ImageNet-9 dataset that can be use to evaluate how much a model is robust to different background.
- For the ablation study in Table 2, do you have the mean/variance across multiple seeds? I just think that the performance of baseline 2, using only local-caption are very close to the ones of AutoGenDA. For most rows, we have a diff of around 0.5% which does not seem to be statistically significant since in Table 1, you had a variance that can be above 3%. So I would be careful about the claim that results are compromised if we do not use AutoGenDA.
- Can you do your the ablations in Table2 with your statistical test in appendix B?
- Typo line 359 on the paragraph title.

---

> ### Author Response · Authors · 2024-11-23
>
> [W1/W4] __More baselines and datasets__. Thanks for your suggestion. Under the limited rebuttal period, we apologize for not being able to include all the suggested datasets and baselines. We add the baseline Feedback-Guided Data Synthesis (FGDS) as suggested, as it is a recent work that also focuses on imbalanced classification, which is more related to our work. We also follow your suggestion to add the CIFAR-10-LT and CIFAR-100-LT experiments. The new results are illustrated in Table 5 of Appendix E. Our method slightly leads the other baselines on the two new datasets.
>
> [W2/W3] __Novelty and relevant work__. We would like to highlight the difference between our work and existing work from two perspectives. (1) Unlike most existing works that study the generation of more diverse and realistic examples on balanced datasets, we focus more on the more challenging imbalanced settings, where some tail classes do not provide sufficient samples to generate diverse augmented images. (2) One large part of our method is an __automated framework__ that automatically learns the effective ratio of different types of augmented samples. This framework can be deployed to any generative method, and we believe such automation is novel in generative augmentation research. We sincerely hope the reviewer can reconsider the contribution of our work by taking into account the two perspectives.
>
> [W4/Q2/Q3] __Ablation study__. In the original ablation study, the results are the average of the 16 imbalanced settings of the four tested datasets. We observe that the reporting of the average may not clearly highlight the edge of our method for more imbalanced scenarios. Therefore, we report all the experimental results in Table 3 of Appendix D. Following your suggestion, we also conduct a statistical test in Table 4 of Appendix D to validate the effectiveness of our method.
>
> [Q1] __Background robustness__. We agree that improving background robustness is one of the major benefits of using AutoGenDA. In addition, we also observe other changes from the generated images, such as the pose and shape of the objects. This can be observed in the newly added augmented samples in Appendix F. This suggests that the variations generated by our method are not limited to background changes.

---

> > ### Comment · Reviewer_AgGB · 2024-11-29
> >
> > Thanks for your response and for also following the suggestions of the other reviewers. I think that the additional baseline and results are adding some strengths to the paper. So, I updated my score.

---

### Official Review · Reviewer_rDKn · 2024-11-01

**Soundness:** 3
**Presentation:** 3
**Contribution:** 2
**Rating:** 6
**Confidence:** 5

**Summary:**

The authors propose a novel generative data augmentation algorithm that captures the class-specific and class-agnostic variances through image captions and integrates these differences into text-guided generation processes by using the captions as text prompts. In order to determine the optimal mixture of real and synthesized images to be included for each class when fine-tuning a classifier on a dataset, the authors additionally propose a novel automated search framework that relies on the Gumble Softmax trick to make the discrete augmentation selection process differentiable and learnable. The experimental evaluation demonstrates the superiority of AutoGenDA  on four object classification datasets under multiple imbalanced and low-shot settings when compared against a few other standard and generative data augmentation baselines.

**Strengths:**

- The paper is well-written and is easy to follow.
- The proposed method outperforms other baselines on four object classification datasets under multiple imbalanced and low-shot settings.
- The experimental section of the paper is elaborate and the ablation study covers several different aspects of AutoGenDA such as the computationally cheaper search-free version, the non class-agnostic and the non class-specific augmentation versions, in Table 2.

**Weaknesses:**

- In Section 1, line 044, the authors mention the ineffectiveness of generative models in learning diverse variations of data-deficient classes in a dataset. However, Section 4.1 - Training mentions that all experiments utilize a pre-trained StableDiffusion 1.4. Since SD1.4 is already pre-trained on a large dataset (LAION-5B [Ref. 1]) spanning several thousands of classes and samples per class, it is fair to assume that the generative model can generalize and capture data diversity even for under-represented classes in the downstream dataset.
- As stated by the authors in the limitations, the method relies on a large image captioning model such as BLIP2 and requires a textual inversion-based fine-tuning of the diffusion model. Both of these are computationally expensive steps and must be performed at inference time, thus hampering AutoGenDA’s applicability. In this light, it is important to compare the computational budgets/throughput time (time taken to generate one image) of AutoGenDA with other baselines included in Section 4.
- It is crucial to include the vanilla text-to-image StableDiffusion 1.4 model as a baseline for comparison in Table 1 and Figure 2. This helps the readers empirically understand the impact of using the proposed methodology vs. simply augmenting the imbalanced classes with a pre-trained generative model.
- How do the authors differentiate class-specific vs. class-agnostic parts of the generated image caption in “a photo of <y> which …..”? For eg: an image-captioning model might generate a caption for an image of a cat as: “a photo of a cat which has thick whiskers and a short tail sitting on a grass field”. In this caption, “thick whiskers”, and “short tail” are class-specific whereas “a grass field” is class-agnostic. If the entire part of the prompt after “a photo of cat which” is chosen for data augmentation of another class y = “cow”, then the StableDiffusion model might end up generating unnatural images.
- The paper misses a few closely related baselines in the related work and for comparison such as [Ref. 2, 3, 4, 5, 6]. Including these in the experimental analyses helps strengthen the authors’ claims.
- There are only 2 qualitative examples of images generated using AutoGenDA in the manuscript in Figure 5. I would encourage the authors to include more generated images of data-deficient classes with different learnt values of selection probabilities $\alpha$. This helps the reader understand the proposed approach intuitively as well as empirically strengthens the authors’ claims.
- Given that the authors specifically aim to demonstrate the efficacy of their method on class-imbalanced datasets, it would be important to include the widely-adopted ImageNet-LT [Ref. 7] dataset for experimental evaluation on long-tailed settings. This helps the reader empirically compare the performance of AutoGenDA with several other methods specifically designed for class-imbalanced datasets.
- The authors include a section (Section 4.3 - Limitations) that highlights a few limitations. However, it is important to also include the computational cost and memory requirements of using a VLM (BLIP2) and fine-tuning a diffusion model (such as StableDiffusion1.4) at inference time for few-shot learning, and even for full-dataset augmentation.


References:

[Ref.1] Schuhmann, Christoph, et al. "Laion-5b: An open large-scale dataset for training next generation image-text models." Advances in Neural Information Processing Systems 35 (2022): 25278-25294.

[Ref. 2] Koohpayegani, Soroush Abbasi, et al. "GeNIe: Generative Hard Negative Images Through Diffusion." arXiv preprint arXiv:2312.02548 (2023).

[Ref. 3] Dunlap, Lisa, et al. "Diversify your vision datasets with automatic diffusion-based augmentation." Advances in neural information processing systems 36 (2023): 79024-79034.

[Ref. 4] Shipard, Jordan, et al. "Diversity is definitely needed: Improving model-agnostic zero-shot classification via stable diffusion." Proceedings of the IEEE/CVF Conference on Computer Vision and Pattern Recognition. 2023.

[Ref. 5] Roy, Aniket, et al. "Cap2aug: Caption guided image to image data augmentation." arXiv preprint arXiv:2212.05404 (2022).

[Ref. 6] Luo, Xue-Jing, et al. "Camdiff: Camouflage image augmentation via diffusion model." arXiv preprint arXiv:2304.05469 (2023).

[Ref. 7] Liu, Z., Miao, Z., Zhan, X., Wang, J., Gong, B., Yu, S.X.: Large-scale long-tailed recognition in an open world. In: CVPR (2019)

**Questions:**

Have the authors considered not fine-tuning the StableDiffusion model and using it out of the box for image-generation using a forward pass? This could be included as an ablation study experiment to understand the importance of fine-tuning to novel downstream classes using textual-inversion.

---

> ### Author Response · Authors · 2024-11-23
>
> [W1/W3/Q1] __Evaluation of vanilla SD1.4__. Thank you for the suggestion. It is true that SD1.4 is pre-trained on a large dataset. However, given a dataset containing less common objects or unseen objects in the pre-training stage, the vanilla SD1.4 may fail to generate good images. Comparison between vanilla Stable Diffusion versus fine-tuned Stable Diffusion model using textual inversion had been studied in the DAFusion work. Under the limited time for the rebuttal period, we did not replicate the comparison on our dataset. However, for the completeness of our paper, we would add this comparison in our final version.
>
> [W2/W8] __Computation overhead__. We add Appendix F to compare the computation overhead of the baseline methods. The computation effort is mainly dominated by the image-generating process. The search effort for AutoGenDA only occupies a small portion of the total computation time. On the CIFAR-10-LT dataset, our method takes 9.6 GPU-hour on a single NVIDIA RTX3090 GPU card. The details can be found in Table 6 of Appendix F.
>
> [W4] __Class-specific vs. class-agnostic__. This is a good question. For local caption images, the caption may contain both class-specific and class-agnostic parts; however, as the captions are extracted in the same class as the input image, the caption should be appropriate. For transfer caption images, the caption can contain class-specific parts unsuitable for the transfer. This highlights the importance of our automated search. In the search phase, the search gradually learns to sample fewer images if the transfer captions lead to unnatural images that harm the classification performance. In addition, we also introduce the filtering mechanism to use captions from more similar classes to mitigate the chance of generating unnatural images.
>
> [W5/W7] __More baselines and datasets__. Thanks for your suggestion. Under the limited rebuttal period, we apologize for not being able to include all datasets and baselines suggested by every reviewer. We add the baseline Feedback-Guided Data Synthesis (FGDS) suggested by Reviewer AgGB as it is a recent work that also focuses on imbalanced classification, which is more related to our work. We also follow the suggestion by Reviewer AgGB to add the CIFAR-10-LT and CIFAR-100-LT experiments. The new results are illustrated in Table 5 of Appendix E. Our method slightly leads the other baselines on the two new datasets.
>
> [W6] __More illustrations__. We add more examples of the generated images in Fig. 11 and Fig. 12 of Appendix H.
>
> [Q1] __Ablation study__. In the ablation study, the results are the average of the 16 imbalanced settings of the four tested datasets. We observe that the reporting of the average may not clearly highlight the edge of our method for more imbalanced scenarios. Therefore, we report all the experimental results in Table 3 of Appendix D. Following the suggestion from Reviewer AgGB, we also conduct a statistical test in Table 4 of Appendix D to validate the effectiveness of our method.

---

> > ### Comment · Reviewer_rDKn · 2024-11-26
> > **Thanks for your response!**
> >
> > Thanks for your response, which already addresses most of my concerns. Thanks also for the extra qualitative and quantitative results, which I feel has improved the draft already to some extent.
> > - I would definitely use the related work suggested to at least improve Section 2.
> > - Please consider adding thee suggestion regarding vanilla SD (and one without fine-tuning) as extra ablation studies to the updated draft. It'll help substantiate your proposition.
> >
> > Looking forward to the updated draft. Thank you!

---

> > > ### Comment · Reviewer_rDKn · 2024-12-02
> > > **follow up responses?**
> > >
> > > Any follow up responses on my remarks, and how those could be potentially addressed? Thanks!

---

> > > > ### Author Response · Authors · 2024-12-02
> > > >
> > > > Thanks for your suggestions. Following your response, we will add the discussions of the suggested related work in section 2 and add the vanilla SD baseline (without textual inversion fine-tuning) for the 16 imbalanced settings to the ablation study. Due to limited time, we haven’t finished running all the additional experiments, but the results will be presented in the camera-ready version of the work. Thank you again for your valuable feedback.

---

> > > ### Comment · Reviewer_rDKn · 2024-12-02
> > > **Thank you**
> > >
> > > I'll raise my score.

---

### Official Review · Reviewer_gN4k · 2024-11-03

**Soundness:** 3
**Presentation:** 4
**Contribution:** 3
**Rating:** 5
**Confidence:** 3

**Summary:**

The paper proposes a method to augment class-imbalanced datasets with pre-trained generative models, where image captions are leveraged to capture the class-specific and class-agnostic variances. A novel search strategy is developed to optimize the data augmentation process for each class. Competitive results are shown on multiple datasets.

**Strengths:**

1. The proposed method uses image captions with the format `a photo of <y> that` to capture the class-specific and class-agnostic variance. Such captions are easy to obtain and straightforward to transfer between different classes.

2. The automated search mechanism can adaptively choose between identity, local-caption, and transfer-caption strategies based on the data distribution of different classes.

3. Results on four datasets show improvement in most settings.

**Weaknesses:**

1. The proposed class-filtering seems limited and lacks ablation studies regarding the parameter $m$ for closest neighbors.
A caption suits classes A and B => A and B must be similar. However, the converse may not hold. As in the last subplot of Figure 5, `sitting on top of the books` may not fit well with the elephant (even though the elephant is a neighbor class of the teddy bear).

2. The search-free baseline is weak in ablation studies. As shown in Fig 4, the optimal strategy from the proposed search method generally assigns a lower probability for augmented samples when the number of samples per class increases. Uniform sampling between the three augmentation types will be suboptimal, even for a fixed strategy. It would also be interesting if the authors could explore the trends of the learned probability w.r.t. N shots.

3. The authors only leverage captions from the dataset itself, which may limit the diversity of class-agnostic features. One can use LLM to generate plausible captions for a class label and augment the images with them.

**Questions:**

1. What dataset is used for the ablation study in Table 2? The numbers shown do not align with those in Table 1.

2. It might be helpful if the authors consider using an LLM to determine whether the caption suits the class. Using the search strategy to assign a lower probability for unsuitable samples such that they affect the classifier less is dodging the issue without addressing it inherently.

3. What are the trends of learned optimal probability when the number of shots increases? Can you provide a possible explanation for the trend?

4. What are the overheads of running the proposed automated search?

---

> ### Author Response · Authors · 2024-11-23
>
> [W1] __Class-filtering__. This is a good question. For local caption images, the captions are extracted in the same class as the input image; the caption should be appropriate. For transfer-caption images, the caption can contain class-specific parts unsuitable for the transfer. This highlights the importance of our automated search. In the search phase, the search gradually learns to sample fewer images if the transfer captions lead to unnatural images that harm the classification performance.  Instead of adjusting the parameter m for each dataset, we tend to fix the parameter and rely on the search algorithm to find the compelling augmented images to be included.
>
> [W2/Q3] __Trend__. We added the analysis on the trend of the learned sampling parameters in Appendix G and Fig. 10. In Fig. 10, we show the averaged augmentation probabilities over all classes for the PASCAL VOC, Caltech, COCO, and LVIS datasets with 2, 4, 8, and 16 shots of data. The learned probabilities for identity images show an increasing trend, while the learned probabilities for local-caption and transfer-caption images show a decreasing trend. This can be explained by the fact that with less data, the classifier easily learns variance within the dataset and demands more varied data to improve generalization performance. Conversely, with more data, the classifier can learn diverse variance information from the real data, thereby assigning a larger probability to sample an identity image.
>
> [W3/Q2] __Use of LLM__. This is a good suggestion. In AutoGenDA, we use the captions contained within the datasets to create augmented samples. We believe the captions contain variations that are specific to the target dataset, thus leading to more customized augmented samples for the dataset. For common objects, we agree that Introducing LLM can create more diverse captions for generating good augmented samples.
>
> [Q1] __Ablation study__. In the ablation study, the results are the average of the 16 imbalanced settings of the four tested datasets. We observe that the reporting of the average may not clearly highlight the edge of our method for more imbalanced scenarios. Therefore, we report all the experimental results in Table 3 of Appendix D. Following the suggestion from Reviewer AgGB, we also conduct a statistical test in Table 4 of Appendix D to validate the effectiveness of our method.
>
> [Q4] __Overhead__. We add Appendix F to compare the computation overhead of the baseline methods. The computation effort is mainly dominated by the image-generating process. The search effort for AutoGenDA only occupies a small portion of the total computation time. On the CIFAR-10-LT dataset, our method takes 9.6 GPU-hour on a single NVIDIA RTX3090 GPU card. The details can be found in Table 6 of Appendix F.

---

### Author Response · Authors · 2024-11-23
**Summary of Changes.**

We would like to thank all the reviewers for their valuable suggestions. Based on the suggestions, we added the following new appendix sections to our work:
- Appendix D reports the full ablation results. We replace the original ablation study (Table 2 in the original paper) with Table 3 in Appendix D. Appendix D also includes a statistical test in Table 4 to validate the effectiveness of our method.
- Appendix E shows additional experiments on long-tail CIFAR-10-LT and CIFAR-100-LT datasets with a new feedback-guided data synthesis (FGDS) baseline as suggested by Reviewer AgGB.
- Appendix F reports and compares the computation overhead of the generative augmentation methods.
- Appendix G provides an additional analysis of the trend of the learned sampling parameters with different numbers of shots of data.
- Appendix H gives more visual examples of the generated augmented data.

The new sections are annotated with the ‘NEW’ tag in the margin space.

---

### Meta-Review · Area_Chair_oEgA · 2024-12-16

**Metareview:**

This paper proposes to augment class-imbalanced datasets by leveraging synthetic data from pre-trained generative models and shows that leveraging such augmentations can improve a classifier's performance. The manuscript was reviewed by four knowledgeable referees. The reviewers acknowledged that the paper tackles an important problem (vKB1) and it is well written and easy to follow  (rDKn, AgGB). The main concerns raised by the reviewers were:
1. the experimental validation, with ablations, baselines, and standard evaluation datasets missing  (gN4k, rDKn, AgGB, vKB1);
2. the use of large scale datasets for pretraining (rDKn, vKB1);
3. the overhead of running the proposed automatic search (gN4k, rDKn).

Moreover, the filtering was not found to be sufficiently justified (gN4k) and there were concerns raised w.r.t. the usage of captions (gN4k).

The authors partially addressed these concerns during the rebuttal and discussion period. For example, the authors introduced new datasets (CIFAR10-LT and 100-LT), some method comparisons, and discussed the computational costs. During the reviewer discussion period, reviewers acknowledged the additional work and results shared by the authors. However, the reviewers remain unconvinced: the concern on the large scale datasets used for pretraining persists, the experimental results remain unconvincing to show the effectiveness of the proposed approach (standard datasets such as ImageNet-LT are missing), and the search strategy does not address the possible incompatibility in transfer captions. Moreover, the significance of the results is not clear, as in many cases results obtained by AutoGenDA appear within standard deviation of the results achieved by competing approaches.

For the above-mentioned reasons, the recommendation after discussion is to reject. The MR agrees with the reviewers' assessment and recommends to reject. The MR encourages the authors to consider the feedback provided by the reviewers to improve future iterations of their work. For example, including ImageNet-LT in the evaluation and providing systematic comparisons with prior art (including all methods and baselines listed by the reviewers) would strengthen the contribution.

**Additional Comments On Reviewer Discussion:**

This is summarized in the meta-review.

---

### Decision · Program_Chairs · 2025-01-22

Reject